# Inhibition of *Fusarium oxysporum* growth in banana by silver nanoparticles: *In vitro* and *in vivo* assays

Natalia Veronica Mendoza[1], Paola Yánez[2], Freddy Magdama[3], Ricardo Pacheco[3], Joel Vielma[1], María Eulalia Vanegas[2], Nina Bogdanchikova[4], Alexey Pestryakov[5], Pablo Chong[3]*

**1** ESPOL Polytechnic University, ESPOL, Facultad de Ciencias Naturales y Matemáticas, Departamento de Ciencias Químicas y Ambientales, Campus Gustavo Galindo, Guayaquil, Ecuador, **2** Center for Environmental Studies (N@NO-CEA Group), Department of Applied Chemistry and Production Systems, Faculty of Chemical Sciences, University of Cuenca, Cuenca, Ecuador, **3** ESPOL Polytechnic University, ESPOL, Centro de Investigaciones Biotecnológicas del Ecuador, Campus Gustavo Galindo, Guayaquil, Ecuador, **4** Centro de Nanociencias y Nanotecnología, Universidad Nacional Autónoma de México, Ensenada, Baja California, México, **5** Research School of Chemistry and Applied Biomedical Sciences, Tomsk Polytechnic University, Tomsk, Russia

* pachong@espol.edu.ec

## Abstract

Fusarium wilt is a devastating disease that affects banana crops worldwide. In Ecuador, bananas are one of the most important commodities and staple food. Nanoparticles are emerging as innovative solutions to control fungal diseases in plant protection. In this study, *in vitro* and *in vivo* assays were carried out to validate *Fusarium oxysporum* growth and disease inhibition. 96-well plates experiments were used to calculate the $IC_{50}$ of three different silver nanoparticle formulations (Argovit-1220, Argovit-1221, and Argovit-C) against four Ecuadorian *Fusarium* strains race 1 (EC15-E-GM1, EC19-LR-GM3, EC35-G-GM6, EC40-M-GM2). More than 95% inhibition rate was obtained at $25\,mg\,L^{-1}$ concentration. Fusarium wilt *in vivo* assay (greenhouse conditions) was carried out with Gros Michel plants, where better control was obtained by applying silver nanoparticles to the roots, reducing disease development by an average of 68%. This study shows that silver nanoparticles have a high antifungal potential for controlling the Fusarium wilt of bananas. To our knowledge, this is the first study to test the potential of AgNPs against *Fusarium oxysporum* race 1 in vitro and in vivo under greenhouse conditions.

## Introduction

Bananas are one of the most important crops worldwide. The fruit is an important staple food and an essential source of nutrition, rich in potassium, vitamins, and dietary fiber [1]. It is also a primary determinant of economic growth, employment, and cultural heritage in many countries. For Ecuador, in particular, bananas are the main exported agricultural product and consumption commodity [2]. The most important emerging threat to the banana industry is the Fusarium wilt caused by the soil-borne pathogen *Fusarium oxysporum* f. sp. *cubense* (Foc) [3]. Foc affects the plant vascular system, causing wilting and yellowing

**Data availability statement:** All relevant data are within the manuscript and its Supporting information files.

**Funding:** Vanegas, E. and Chong P., received Grant FIASA-CA-2023-008 from FIASA - Fondo de Investigación para la Agrobiodiversidad, semillas y Agricultura Sustentable", research. https://www.iniap.gob.ec/el-fiasa-un-me-canismo-financiero-para-el-desarrollo-e-in-novacion-de-una-agricultura-sustentable/ Bogdanchikova N. received Grant 22-13-20032 from Russian Science Foundation and Tomsk Region. https://rscf.ru/en/news/en-57/rus-sian-science-foundation-released-2023-results/ The funders had no role in study design, data collection and analysis, decision to publish, or preparation of the manuscript.

**Competing interests:** The authors have declared that no competing interests exist.

that eventually leads to the plant death [4]. In the 1960s Foc race 1 cost significant losses to the banana industry and a profound shift in the way bananas were cultivated [5]. The main change was the replacement of the Gros Michael cultivar with varieties belonging to the Cavendish subgroup resistant to Foc race 1. Currently, the banana industry is threatened again by another race of Foc, colloquially known as tropical race 4 (TR4), and capable of causing disease in more than 60% of all varieties of banana in the world, including Cavendish cultivars [5]. The monoculture of bananas, its genetic uniformity, the non-availability of chemical control, and the lack of proper phytosanitary national policies guarantee the failure of disease management, biosafety, and quarantine procedures [3]. Although a new recombinant banana *Fusarium*-resistant cultivar has been liberated in Australia [6], it is not broadly available, and some countries like Ecuador have imposed legal restrictions on the adoption of certain technologies, including recombinant pathogen-resistant banana varieties [7,8], leaving no other effective control measures for this disease. These reasons emphasize the urgency to develop new solutions for *Fusarium* control in banana plantations.

New alternative technologies, such as nanotechnologies are rising for multiple biological applications. Based on their antimicrobial properties nanoparticles (NPs) have been widely used in many agricultural applications [9]. The advantages of the NPs for plant pathogens control are their low-dose effectiveness [9]. Silver (Ag) and its derivatives are acknowledged for their antimicrobial properties [10,11]. Silver nanoparticle (AgNP) modes of action include the disruption of the microbe cell membrane potential [10] or the inhibition of processes like DNA or RNA synthesis, ultimately leading to cell death [12]. The effectiveness of AgNPs as antimicrobial agents has been extensively reported in the literature [13]. AgNPs have been proven to be effective against *Botrytis cinerea*, *Alternaria alternata*, *Monilinia fruticola*, *Colletotrichun gloeosporioides*, and *Fusarium oxysporum* f. sp. *Radices-Lycopercisi*, *Fusarium solani* and *Verticillium dahlia* on both *in vitro* and *in vivo* tests [11]. In particular, AgNPs have been shown to be particularly lethal at low doses to different *Fusarium* species [11]. Mahdizadeh et al. (2015) studied AgNPs against many pathogenic and beneficial fungi, showing a differential dose effect for each species, with *Trichoderma harzianum* the least affected. In this case, Mahdizadeh et al. (2015) argue that the beneficial fungi *T. harzianum* could be protected using a proper dose that will control the pathogens and simultaneously have a milder effect on the beneficial fungi growth [10]. On the other hand, response variability to AgNPs among strains in nature or laboratory settings was expected [14]. Previous studies showed that fungi populations, including species like *Candida, Aspergillus, Cryptococcus*, and *Pneumocystis*, have different susceptibility rates to AgNP antifungal drugs [14,15]. It is important to prevent the development of resistance in the fungal population, whether by overdose or overuse of the compound, to avoid the selection of low susceptible strains.

In the present study, our main goal was to assess the inhibitory effect of AgNPs over Fusarium wilt on bananas. Our study shows *in vitro* control over the pathogen growth at AgNP doses as low as 25 mg L$^{-1}$ and a substantial reduction of the disease severity in a greenhouse experiment with just one AgNP application at 100 mg L$^{-1}$.

## Materials and methods

### Fungal strains

*Fusarium oxysporum* f. sp. *cubense* (Foc) race 1 strains EC15-E-GM1, EC19-LR-GM3, EC35-G-GM6, and EC40-M-GM2 all of them belonging to VCG-0120 were provided by the collections of the Biotechnology Research Centre of Ecuador (CIBE) from ESPOL Polytechnic University [3].

## Culture conditions for in vitro assays

Strains were cultivated from the stock water suspensions on potato dextrose agar (PDA) and incubated at 28°C for 3 days. Cultures grown on PDA were covered with 5–9 mL of Tween 20 (0.05%), and the inoculum suspension was harvested by a gentle but exhaustive surface scraping with a sterile loop. Then, the obtained solution was transferred to a 50 mL conical tube. Aliquots of the solution obtained at 1:10 and 1:100 were prepared and quantified under the microscope cell-counting with a Neubauer chamber.

## AgNPs synthesis and formulations

All three AgNP samples, codes Argovit-1220, Argovit-1221, and Argovit-C, were lots based on Argovit™ commercial formulations (Research and Production Center "Vector-Vita" Ltd, Novosibirsk, Russia). Argovit™ is a water suspension of AgNPs. The metallic content of Ag is 12 mg mL$^{-1}$ (1.2 wt. %) with 188 mg mL$^{-1}$ (18.8 wt. %) of the stabilizer with a final concentration of AgNPs 200 mg mL$^{-1}$ (20 wt. %) suspension. All stocks and aliquots were stored in the dark at 4°C. The characterization of the AgNP samples is shown in Table 1.

## Fungal in vitro growth inhibition

A 96 microwell plate was used to perform growth inhibition bioassays of Foc race 1 with AgNPs. The antifungal properties of AgNPs were reproduced on three AgNP samples and at four strains of Foc. The total volume of each well was 100 µL, which was comprised of 75 µL of Mueller Hinton Broth (Titan Biotech Ltd., Rajasthan, India) medium, 15 µL of AgNPs, and 10 µl of each fungal strain. The medium was dispensed, and the nanoparticles were dosed at different concentrations; 0, 0.8, 1.6, 3.1, 6.3, 12.5, 25, 50, and 100 mg L$^{-1}$ calculated for metallic Ag with stabilizer. Each strain was added at a concentration of $1 \times 10^4$ spores per millilitre. The plate was incubated for 3 days at 28°C to ensure optimal fungus growth. The compound alamarBlue™ HS Cell Viability Reagent (Thermo Fisher Scientific Inc., Waltham, Massachusetts, USA) was added aseptically at 10% of the total well volume, to measure the cell viability for each treatment and determine its half-maximal inhibitory concentration (IC$_{50}$). The study was performed three times, and each treatment included three replications.

The percentage of inhibition was evaluated by measuring cell viability with the alamarBlue™ compound. In this reaction, the molecule resazurin (non-fluorescent blue colour) is reduced to resorufin (fluorescent pink colour) [17]. Then the percentage of inhibition was determined according to the manufacturer's protocol [17]. IC$_{50}$ was calculated to determine the AgNP

**Table 1. Characteristics of AgNP samples studied in the present work.**

| Sample | Stabilizer type | Content, wt. % | | AgNP characteristics according to HRTEM data[*] | |
| --- | --- | --- | --- | --- | --- |
| | | Ag | Stabilizer | The average diameter, nm | Morphology |
| Argovit-1220 | Polyvinylpyrrolidone (PVP) with glucose additive | 1.2 | 18.8 | Bimodal distribution with maxima at 8 nm and 80 nm | 8 nm spheroidal, 80 nm pyramidal |
| Argovit-1221 | Hydrolyzed protein (HP) | 1.2 | 18.8 | 8.5 ± 3.3 | Spheroidal |
| Argovit-C | Mixture of PVP (1/3) and HP (2/3) | 1.2 | 18.8 | 14.95 ± 10.1 | Spheroidal |

[*]HRTEM data was taken from the previous publication of our group [16].

antifungal potential, and the GraphPad Prism program version 10.0.2 (Home – GraphPad Software, Boston, Massachusetts USA, www.graphpad.com) was used for this purpose.

## Preparation of the inoculum for greenhouse bioassays

In a 1 L flask, a solution with 700 mL of filtered water and 5.6 g of mung bean was placed and autoclaved for 15 min at 121 °C with aluminium foil. Once ready, the solution was kept at room temperature until its temperature was reduced to approximately 40 °C. Five mycelium pieces (0.5 x 0.5 cm) of Foc race 1 were placed in the solution and the flask nozzle was sealed with cotton plugs. The solution with the pieces of mycelium was incubated for 8 days at 120 RPM and 25 ± 2 °C. Once the incubation period was over, the solution was filtered using a vacuum pump and a double layer of sieving gauze. The filtrated solution with spores was stored at 4 °C until its later use. A diluted spore solution at 1:100 was used three times to estimate spore concentration with a Neubauer cell counting chamber (Boeco, Hamburg, Germany). The final solution was adjusted to $1 \times 10^4$ spores per millilitre.

## Greenhouse bioassay conditions

Under greenhouse conditions, 84 three-month-old Gros Michel banana plants were set on a substrate composed of compost, sand, and rice husks in a 6:2:2 ratio, for the *in vivo* bioassays. Two experiments were conducted using three AgNP samples (Argovit-1220, Argovit-1221, and Argovit-C) at 100 mg L$^{-1}$; applied through both foliar and drench methods. Two inoculation methods were employed: drenching and root dipping. Positive and negative controls were included to compare with the treatments. The positive controls in the experiment consisted of plants that had been infected with Foc race 1 without any AgNPs treatment, while the negative controls were plants without inoculation but with the AgNPs excipient at the same concentration (100 mg L$^{-1}$). Each treatment had five replicates, along with six negative controls for each application method and six positive controls for each inoculation method in the bioassays.

For inoculation by drenching, 100 mL of a concentrated inoculum solution at $2 \times 10^5$ spores per millilitre was poured into the substrate. Expanded polystyrene plates were placed under the pots to keep the spore solution in contact with the substrate. For the inoculation by root dipping, the roots were separated from the substrate to be soaked in the inoculum solution for 30–40 min. For this method, we decided not to cut the roots as most methodologies do, to not stress the plant in this way, and not compromise the results of the experiment. Once this period ended, the plants with the infected roots were planted again in the fertilized substrate.

After 72 h of inoculation, the plants were watered, and AgNPs were applied to the leaves and the roots. The plants were kept under evaluation for 40 days, then harvested, and photos of corm damage were taken per plant for analysis.

## Evaluation of antifungal activity

The software ImageJ was used to quantify the corm-affected area by Foc race1 following the methodology of Pride et al. (2020) [18]. The selection of the corm-healthy and necrotic tissue area was obtained by adjusting and changing the colour threshold parameters (hue, saturation, and brightness) to measure each area. The following formula described by Islam and Islam (2015) [19] was used to calculate the percentage of the infected area in each plant:

$$\%A_I = \frac{A_R}{A_T} \; x \; 100 \; \%,$$

where $A_R$ is the area of the affected region of the corm, and $A_T$ is the total area of the corm.

## Statistical analysis

The data in this study was analyzed based on a comparison with the treatments and their respective control data, using ANOVA. Statistical significance difference was considered when p values were less than 0.05. All the tests were performed with the software InfoStat (version 2008, F.C.A-U.N.C, Argentina). After the analysis, the data were $\log_2$-transformed to show a better visual scale representation in some figures.

## Results

### In vitro bioassays of Fusarium interaction with AgNPs

The results obtained from inhibition tests performed *in vitro* showed that AgNPs have an antifungal effect against Foc race 1 strains. Fig 1 presents the mycelial growth inhibition percentage obtained in the experiment for each treatment tested. All three AgNP samples show antifungal activity against all the strains at different AgNP concentrations. Most strains reached approximately 50% fungal growth inhibition at concentrations around 3.1 mg $L^{-1}$ and 6.3 mg $L^{-1}$. More than 90% inhibition was achieved at concentrations from 25 mg $L^{-1}$ to 50 mg $L^{-1}$. The study strains showed different sensitivities to the AgNPs. The strain Ec35-G-GM6 showed the lowest sensitivity toward the AgNP samples (S1 Table).

Fig 2 shows the calculated $IC_{50}$ values from the strains´ responses. The $IC_{50}$ values from the interaction of Argovit-1220 with EC15-E-GM1, EC19-LR-GM3, and EC40-M-GM2 strains were in the range of 3.00 mg $L^{-1}$–3.75 mg $L^{-1}$, and the $IC_{50}$ value for the EC35-G-GM6 strain was a value of 7.39 mg $L^{-1}$. For the Argovit-1221 sample, the $IC_{50}$ values with EC15-E-GM1, EC19-LR-GM3, and EC40-M-GM2 were from 4.49 mg $L^{-1}$–6.66 mg $L^{-1}$, and for the EC35-G-GM6 strain, the $IC_{50}$ was almost triple (23.26 mg $L^{-1}$). For Argovit-C, the $IC_{50}$ for EC15-E-GM1, EC19-LR-GM3, and EC40-M-GM2 varied from 1.35 mg $L^{-1}$ to 2.12 mg $L^{-1}$, again for the EC35-G-GM6 strain required a higher concentration (4.23 mg $L^{-1}$).

Fig 3 shows that EC35-G-GM6 demonstrated the highest average concentration of $IC_{50}$ (11.63 ± 9.47 mg $L^{-1}$), indicating a significant difference between the rest of the treatments. However, strains EC15-E-GM1, EC19-LR-GM3, and EC40-M-GM2 showed not statistically different between their $IC_{50}$ values.

In Fig 4, on average, Argovit-1221 appeared as the least effective AgNP sample as it had a higher $IC_{50}$ value (10.21 ± 8.44 mg $L^{-1}$) than the other samples, showing a statistical difference with Argovit-1220 and Argovit-C values. However, there is no statistical difference between Argovit-1220 (4.33 ± 2.26 mg $L^{-1}$) and Argovit-C (2.52 ± 1.13 mg $L^{-1}$).

### Greenhouse bioassays of Fusarium interaction with AgNPs

The percentage of disease inhibition was determined by considering the ratio of the infected area to the total area of the corm. Table 2 presents the results obtained with the radicular and foliar application of AgNPs for the two Foc inoculation methods (root dipping and drenching). The three AgNP samples showed good control for both modes of application. The range of inhibition obtained by the Argovit-1220 sample was from 64% to 83%. Likewise, the Argovit-C inhibition range was from 58% to 78%. The Argovit-1221 range of inhibition obtained was from 35% to 78%. Overall, the Argovit-1220 treatment showed a higher control than the other AgNPs-treated groups with an inhibition average of 73.96% ± 9.44. The Argovit-C sample exhibited a similar control than Argvoti-1220 with an inhibition average of 68.47% ± 9.33. The Argovit-1221 was the least effective in controlling the pathogen with an inhibition average of 59.38% ± 18.46.

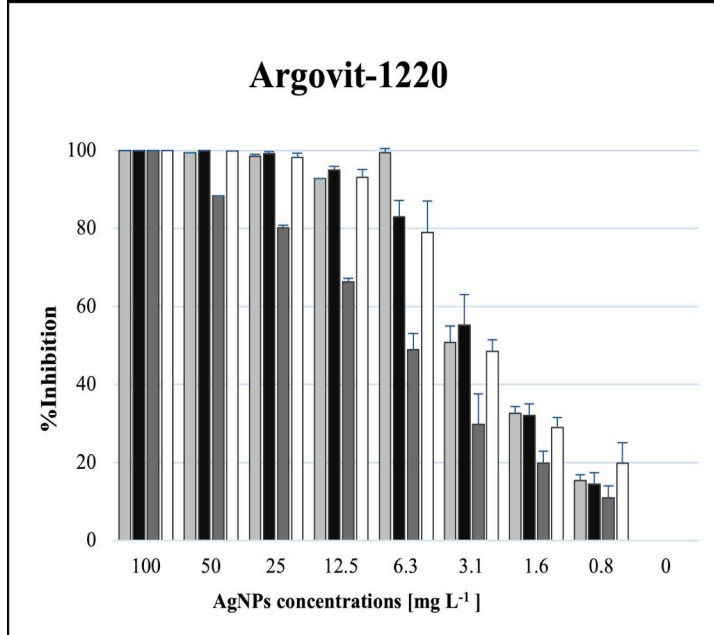

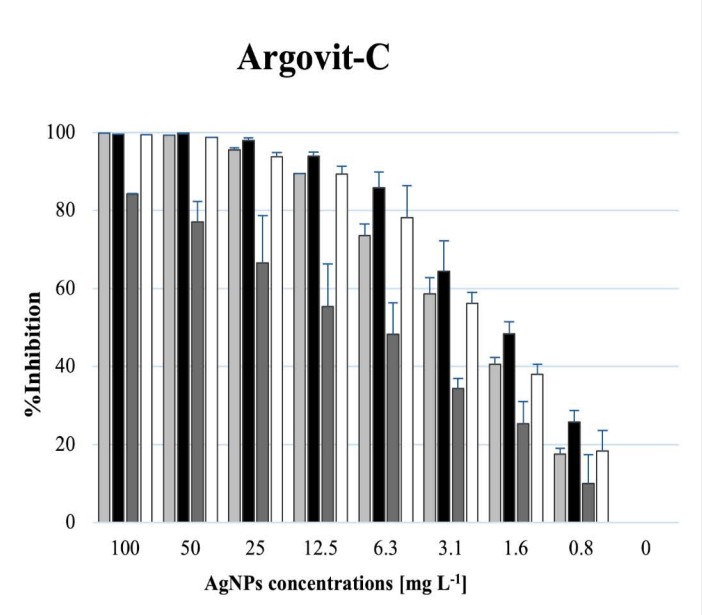

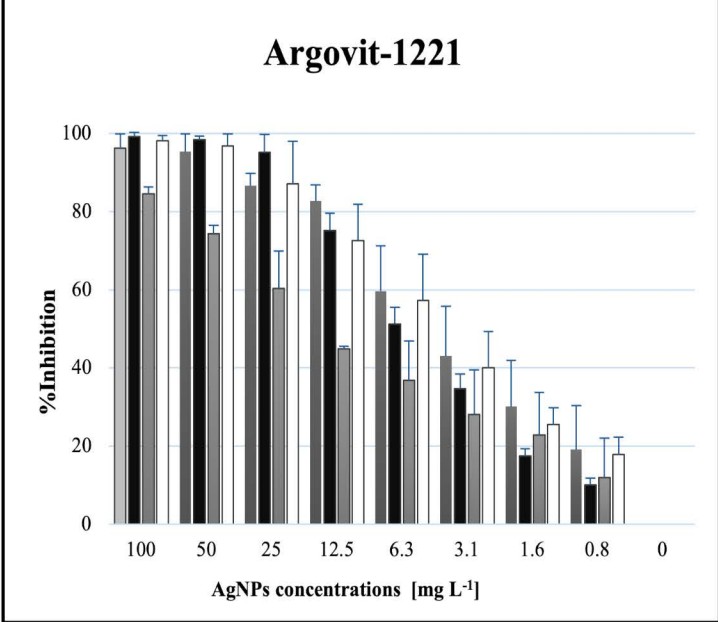

**Fig 1. Growth inhibition of Foc race 1 at different AgNP concentrations for three AgNP samples.** Bar diagrams show the effect of the AgNP samples on Fusarium inhibition growth.

The relation to Table 2, Figs 5 and 6 provides a visual representation of the corm condition, with a comparison between the AgNPs treatment and their respective controls (positive and negative). The photos in Fig 5 show the results of the foliar application of AgNPs, and Fig 6 shows the results of the radicular application. The photographs exhibited that the Fusarium wilt was controlled and held back by the three samples of silver nanoparticles.

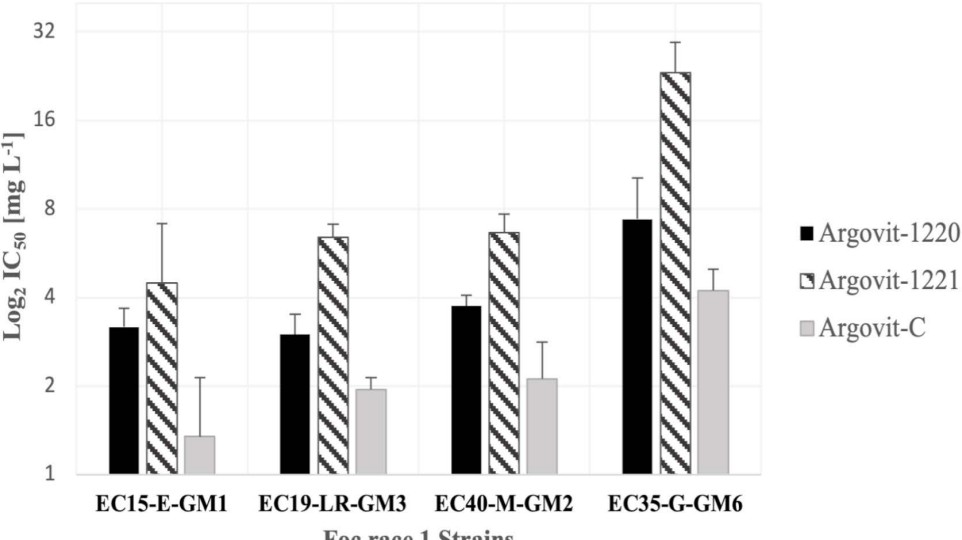

**Fig 2. Means IC$_{50}$ (Log$_2$) values of the AgNP samples.** Values of the half inhibitory concentrations for the Argovit-1220, Argovit-1221, and Argovit-C interacting with Foc race 1 strain: EC15-E-GM1, EC19-LR-GM3, EC35-G-GM6, and EC40-M-GM2.

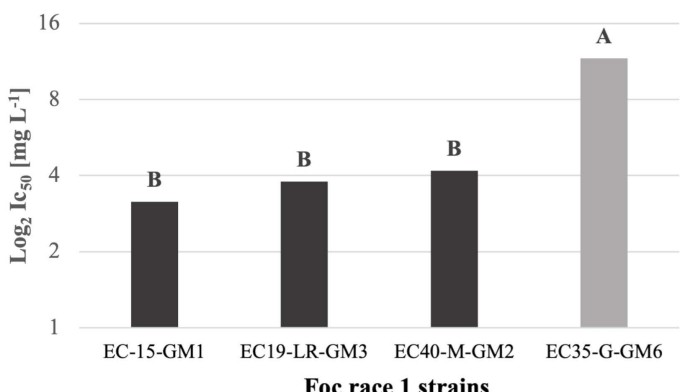

**Fig 3. IC$_{50}$ (Log$_2$) combined means values of the different Foc race 1 strains.** The letters A and B represent the statistically significant difference between them according to the ANOVA test (p > 0.05).

## Discussion

### Antifungal activity in vitro

AgNPs possess unique physicochemical properties that contribute to their antimicrobial activity [13]. These include a high surface area-to-volume ratio and the ability to generate reactive oxygen species (ROS), which can lead to oxidative stress and subsequent cell death in microorganisms [20]. A study by Terzioğlu et al. [14] revealed that AgNPs exert their antifungal effects by inactivating vital enzymes, increasing ROS production, and reducing mitochondrial membrane potential, ultimately resulting in fungal cell death. Another reported mechanism is the diffusion of AgNPs into the fungal cells, where they disrupt the cell membrane [10]. A recent study from China confirmed these mechanisms in *Fusarium oxysporum* (Schl.) f.sp. *melonis*, demonstrating

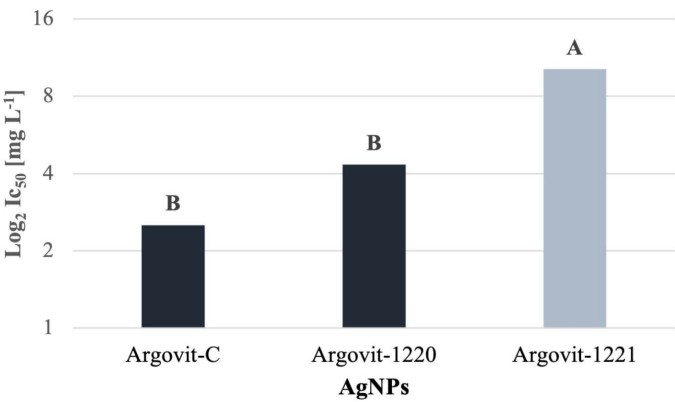

**Fig 4. Comparison between the combined average IC$_{50}$ (Log$_2$) values of the AgNP samples for all Foc race 1 strains.** The letters (A, B) mean the significant difference between them according to the ANOVA test (p > 0.05).

that AgNPs not only increase ROS production and cause cell wall and membrane damage, but also interfere with key metabolic processes [21]. This disruption affects enzyme activities such as catalase (CAT), superoxide dismutase (SOD), and peroxidase (POD), reducing the fungus's ability to defend against oxidative damage. Furthermore, AgNPs were found to cause lipid peroxidation, protein denaturation, and organelle damage, leading to fungal cell death [21].

A previous study on *Fusarium oxysporum* f. sp. *radicis-lycopersici* showed that AgNPs control Foc at a concentration of 25 mg L$^{-1}$ and higher [20]. In our experiments, analysis of the IC$_{50}$ values revealed notable differences in susceptibility to AgNPs among the tested Foc race 1 strains. Overall, lower IC$_{50}$ values indicated higher potency, showing that lower concentrations of AgNPs were required to inhibit fungal growth by 50%. The nanoparticles that appear to have the most promising antifungal potential are the Argovit-1220 and Argovit-C samples, as they require lower concentrations to achieve effective inhibition (12.5 mg L$^{-1}$–25 mg L$^{-1}$). In contrast, Argovit-1221 required higher concentrations (50 mg L$^{-1}$–100 mg L$^{-1}$), as shown in S1 Table.

Strain EC-35 exhibited the highest IC$_{50}$ values across all three tested samples of AgNPs, indicating that it had the lowest susceptibility to the AgNPs compared to the other strains tested. This highlights that each strain has a distinct sensitivity to the treatments. The

**Table 2. Results of the means values of Foc race 1 interaction with each AgNP in the greenhouse bioassay.**

| AgNPs | Inoculation method | Application type | Necrotic area, % | Inhibition, % |
|---|---|---|---|---|
| Argovit-1220 | Root Dipping | Radicular | 4.08% ± 0.01 | 81.29% ± 0.04 |
| Argovit-1221 | Root Dipping | Radicular | 6.67% ± 0.03 | 69.39% ± 0.13 |
| Argovit-C | Root Dipping | Radicular | 5.51% ± 0.02 | 74.73% ± 0.08 |
| Argovit-1220 | Drench | Radicular | 12.71% ± 0.06 | 67.05% ± 0.16 |
| Argovit-1221 | Drench | Radicular | 17.31% ± 0.02 | 55.10% ± 0.06 |
| Argovit-C | Drench | Radicular | 14.22% ± 0.04 | 63.12% ± 0.10 |
| Argovit-1220 | Root Dipping | Foliar | 3.73% ± 0.02 | 82.86% ± 0.1 |
| Argovit-1221 | Root Dipping | Foliar | 4.88% ± 0.04 | 77.57% ± 0.17 |
| Argovit-C | Root Dipping | Foliar | 4.82% ± 0.02 | 77.85% ± 0.1 |
| Argovit-1220 | Drench | Foliar | 8.00% ± 0.02 | 64.64% ± 0.07 |
| Argovit-1221 | Drench | Foliar | 14.62% ± 0.01 | 35.44% ± 0.4 |
| Argovit-C | Drench | Foliar | 9.47% ± 0.01 | 58.19% ± 0.04 |

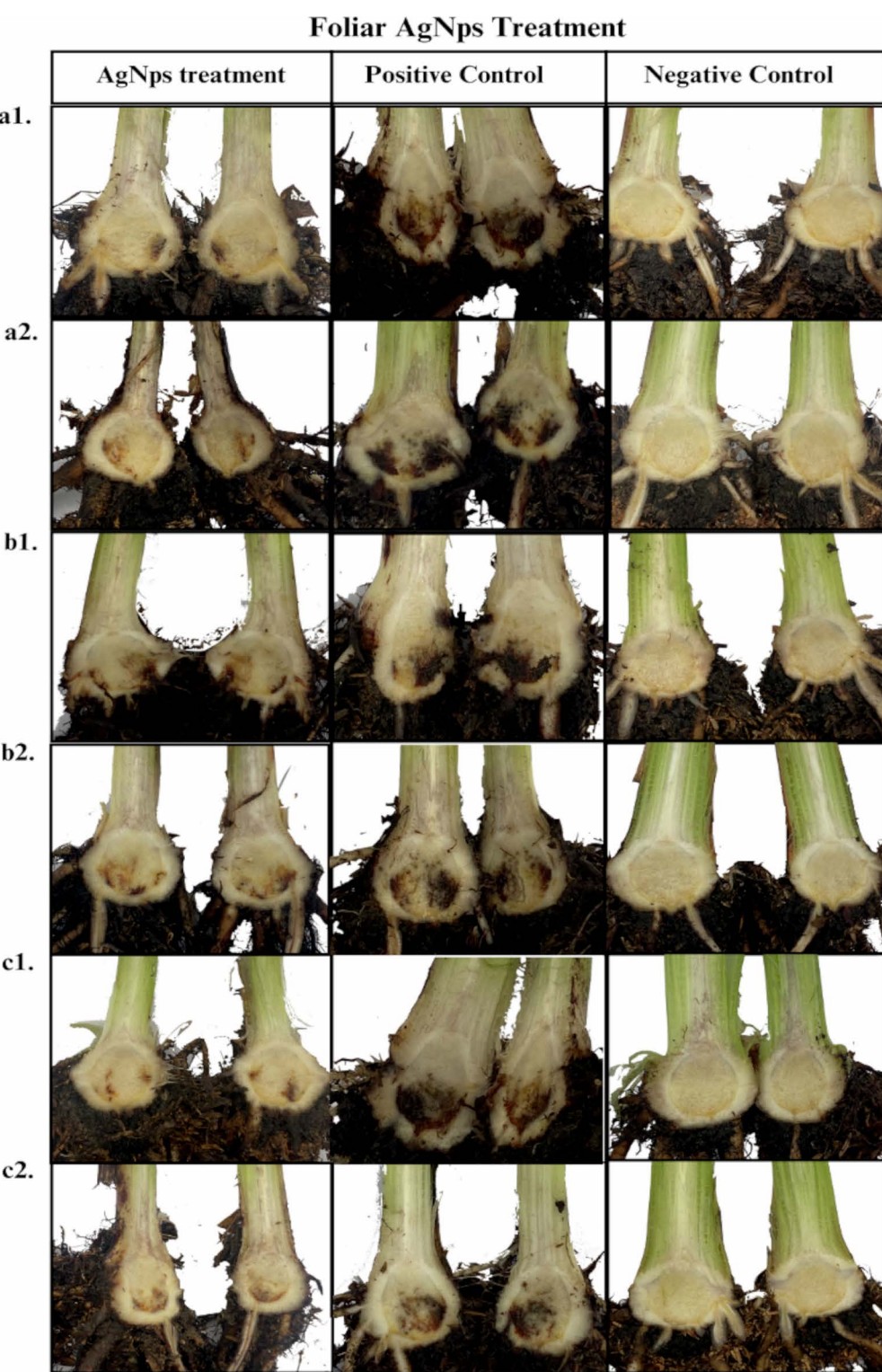

**Fig 5. Pictures of the control of Fusarium wilt by Foc race 1 in Gros Michel plants with foliar AgNPs application.**
a1. Argovit-1220 with root dipping inoculation, a2. Argovit-1220 with drench inoculation, b1. Argovit-1221 with root dipping inoculation, b2. Argovit-1221 with drench inoculation, c1. Argovit-C with root dipping inoculation, and c2. Argovit-C with drench inoculation. Positive control (no AgNP application to inoculated plants) and negative control (not inoculated plants without AgNP application).

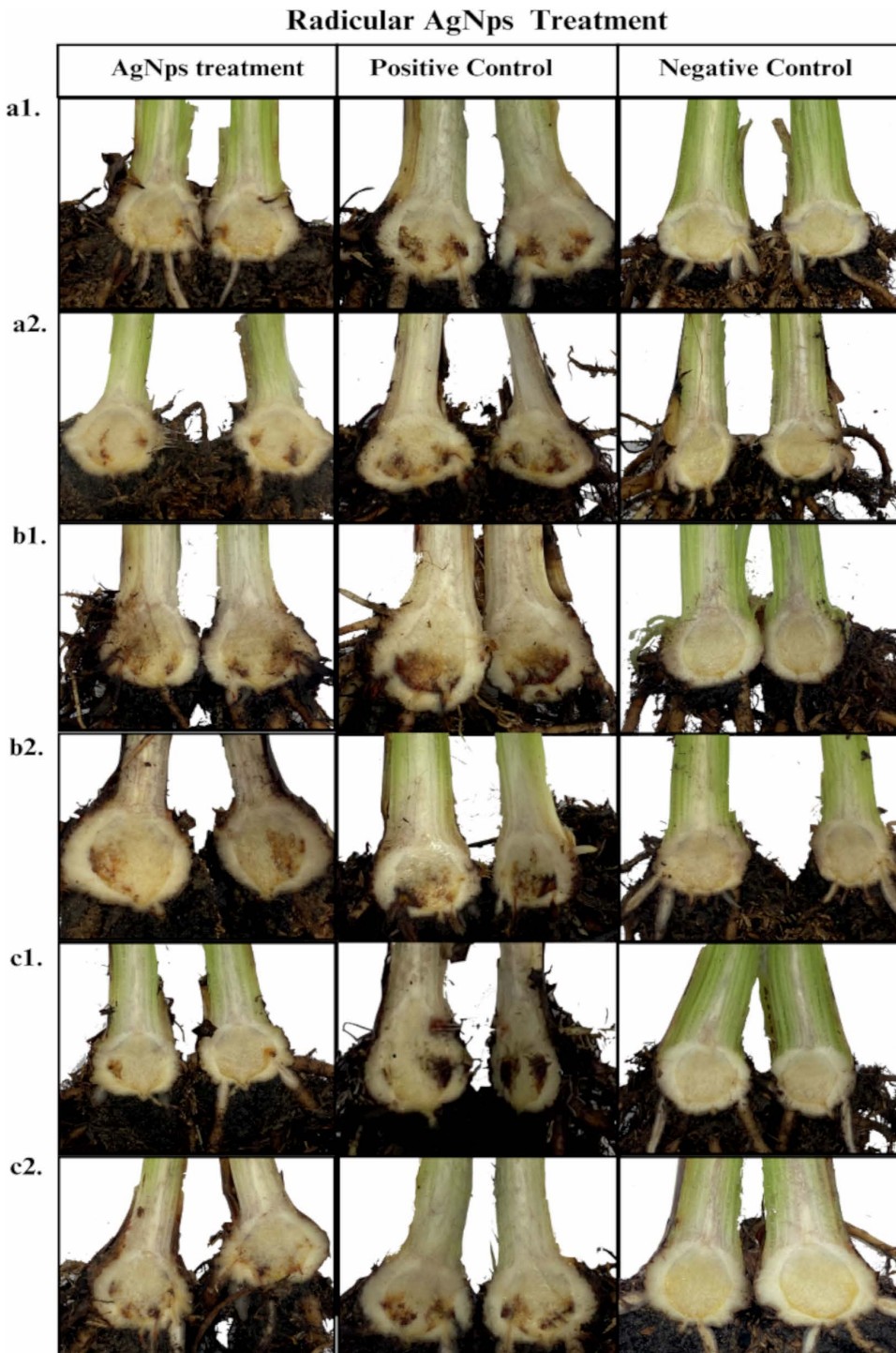

**Fig 6. Control of Fusarium wilt by Foc race 1 in Gros Michel plants with AgNps radicular (drench) application.**
a1. Argovit-1220 with root dipping inoculation. a2. Argovit-1220 with drench inoculation. b1. Argovit-1221 with root dipping inoculation. b2. Argovit-1221 with drench inoculation. c1. Argovit-C with root dipping inoculation and c2. Argovit-C with drench inoculation. Positive control (no AgNP application to inoculated plants) and negative control (not inoculated plants without AgNP application).

variation in susceptibility to AgNPs among different Foc strains underscores the importance of considering strain-specific responses when evaluating antifungal agents. Some strains may exhibit reduced susceptibility to antifungal agents due to mutations, innate defence, or stress mechanisms. The study performed by Terzioğlu et al. [14] revealed that fungi respond to heavy metals, including silver, through a variety of mechanisms, including sequestration, efflux facilitation, and reduction of influx [14].

The concentrations of AgNPs that have been reported to inhibit *Fusarium* species vary across different studies. For instance, Singh et al. (2019) observed complete inhibition of *F. oxysporum* at $75\,mg\,L^{-1}$ of AgNPs [22], while Akpinar et al. (2021) demonstrated effective inhibition of *Fusarium oxysporum* f. sp. *radicis-lycopersici* strains at concentrations between 25 to $50\,mg\,L^{-1}$ [20]. A study from China on *Fusarium oxysporum* (Schl.) f.sp. *melonis*, reported inhibition on PDA plates at concentrations ranging from $100{-}200\,mg\,L^{-1}$ [21]. These variation in inhibitory concentration may be influence by factors such as AgNPs size, which has been shown to affect antifungal efficacy [20,23]. Higher concentrations of AgNPs can impact their antifungal activity, with higher concentrations often leading to increased inhibition of fungal growth [12,24].

In the present study, we found that at concentrations between $25\,mg\,L^{-1}$ and $50\,mg\,L^{-1}$ achieved over 90% inhibition of fungal growth, suggesting that AgNPs at these levels could serve as a promising solution for controlling Fusarium wilt caused by Foc race 1 in banana crops. Additionally, other studies confirm that the fungus does not quickly develop resistance to AgNPs [21], unlike traditional fungicides that often lead to resistant populations. These further underscores the potential of AgNPs as an effective and sustainable alternative for managing banana Fusarium wilt.

## Antifungal activity of AgNPs in greenhouse conditions

The antifungal potential of the three types of AgNPs (Argovit-1220, Argovit-1221, and Argovit-C) was evaluated against Fusarium wilt under greenhouse conditions using two AgNPs application: foliar and radicular, along with two methods of Fusarium inoculation: root dipping and drenching. At a concentration of $100\,mg\,L^{-1}$, both types of applications significantly reduced Foc race 1 infection, as demonstrated by the marked difference between the treatment and the positive controls in Figs 5 and 6. Argovit-1220 had the highest antifungal potential, followed by Argovit-C, which also showed a similar control. The least effective, according to the results, was the Argovit-1221.

The inoculation methods also influenced disease progression. Drenching inoculation resulted in a higher infection rate on the corm compared to root dipping, despite that previous studies have reported better infection results with the root dipping method [25–27]. This difference could be explained by the fact that when using the root dipping inoculation method, many studies have made 2 cm root cuts and stressed the plant. Hence, the plant root wounds come in direct contact with the sporulated solution, making it easier for *Fusarium* spores to enter and colonize the root vascular tissue. However, the drenching method involved a longer contact time with the solution due to the use of polystyrene plates under the pots to retain the solution for a longer period.

Both the radicular application and foliar application were effective as antifungal agents. This means that it would be possible and easy to apply the AgNPs in the banana farms by the irrigation system and aerial spraying. This is a positive result because even though radicular application offers the advantage of direct contact between the AgNPs with the soilborne Fusarium, foliar application is also considered a more suitable method for fungicide application in agricultural settings. Additionally, applying AgNPs directly to the leaf, rather than the root, can help reduce undesirable effects on soil microorganisms and the environment [28]. In practice, both foliar and root application methods can be integrated to enhance efficacy.

Some studies report that AgNPs may act as a bio-stimulant, enhancing the plant's defense mechanisms against fungal infections by triggering the production of ROS and activating various stress-related pathways. This activation can strengthen the plant's defense response, making it more resistant to pathogen attacks [29–31]. In the context of this study, the bio-stimulant properties of AgNPs suggest that, beyond direct antifungal effects, AgNPs may also contribute to an overall increase in the plant's resilience, providing a dual benefit: inhibiting fungal growth and fortifying the plant's natural defense systems.

The method of synthesis of AgNPs also plays an important role in their antifungal potential [32]. While the exact mechanisms are not fully understood, AgNPs are detrimental to fungal cell growth [33]. In general, the interaction of AgNPs with fungal cells has shown good controls for plant pathogenic fungi, suggesting their potential as an alternative to conventional fungicides [34]. The novelty of this study is our *in vivo* results that are not comparable with other literature data because this work is the first *in vivo* assay under greenhouse conditions of the use of AgNPs against *Fusarium oxysporum* f. sp. *cubense.*

These results show that AgNPs have a strong potential to control Fusarium wilt and could be used in agricultural practice. However, while these results are encouraging, several studies have raised concerns about the biosafety of AgNPs [35,36], particularly regarding their environmental impact [37]. One of the main concerns is phytotoxicity. At high concentrations, AgNPs can inhibit seed germination, reduce root growth, and lower biomass in crops like rice, maize, and beans, which limits their broader application [36]. Furthermore, AgNPs may negatively affect non-target organisms such as beneficial soil microbes, disrupting essential processes like nutrient cycling and soil health. Additionally, the potential for environmental dispersion raises concerns, as AgNPs could accumulate in water systems and air through surface runoff and leaching, leading to long-term ecological disruption. More studies should be done to analyse the impact of AgNP application on the environment in agricultural settings.

## Conclusion

The bioassay results demonstrated the promising antifungal activity of studied AgNP samples (Argovit-1220, Argovit-1221, and Argovit-C) against Foc race 1 strains, as shown by the low $IC_{50}$ values obtained. This is the first study demonstrating the systemic antifungal activity of AgNPs against Foc race 1 on banana plants. These results are relevant for the development of additional fungicides to control soilborne pathogens, as currently there is only one conventional fungicide proposed against Fusarium wilt [38]. This study underscores the importance of selecting appropriate application methods to maximize AgNP efficacy in controlling Foc infections. Further research is needed to optimize AgNP formulations and application protocols for field-scale implementation, and the impact of AgNPs on plant endophytes and the environment. This study contributes valuable insights into the development of innovative pesticide agents for managing fungal diseases in agriculture.

## Supporting information

**S1 Table. Fungal growth inhibition percentages of *Fusarium oxysporum* strains by different concentrations of AgNPs.** Inhibition percentages of nine different concentrations of AgNPs against four FOC race 1 strains. The table presents the inhibitory effects observed for each FOC strain at AgNP concentrations ranging from $0\,mg\,L^{-1}$ to $100\,mg\,L^{-1}$. Values represent the mean percentage inhibition ± standard deviation from three independent experiments with three technical replicates.
(PDF)

**S1 Data. Data in vitro, data in vivo and IC-50 in vitro.**
(7Z)

## Acknowledgments

We extend our sincere gratitude to the Centro de Investigación de Biotecnología del Ecuador (CIBE) for supplying the strains used in our experiments and for granting access to their facilities, which were essential for conducting the assays. We thank the International Network of Biotechnology with Impact on Biomedicine, Food, and Biosafety for their academic support and for providing the nanoparticles used in this study. Additionally, we would like to acknowledge the insightful contributions of Evelin Odalis Escobar Hidalgo and Freddy Roger Carlos García, graduate students of Chemical Engineering at ESPOL, and Aracely Paguay Salcan, our colleague from CIBE, whose input was instrumental in shaping this work.

## Author contributions

**Data curation:** Natalia Veronica Mendoza, Pablo Chong.

**Formal analysis:** Natalia Veronica Mendoza, Paola Yánez, Pablo Chong.

**Funding acquisition:** María Eulalia Vanegas, Nina Bogdanchikova, Pablo Chong.

**Investigation:** Natalia Veronica Mendoza, Paola Yánez, Freddy Magdama, Ricardo Pacheco, Joel Vielma, María Eulalia Vanegas, Alexey Pestryakov, Nina Bogdanchikova, Pablo Chong.

**Methodology:** Natalia Veronica Mendoza, Paola Yánez, Freddy Magdama, Ricardo Pacheco, Joel Vielma, María Eulalia Vanegas, Alexey Pestryakov, Nina Bogdanchikova, Pablo Chong.

**Project administration:** Pablo Chong.

**Resources:** Freddy Magdama, Joel Vielma, María Eulalia Vanegas, Alexey Pestryakov, Nina Bogdanchikova, Pablo Chong.

**Supervision:** Freddy Magdama, Joel Vielma, María Eulalia Vanegas, Nina Bogdanchikova, Pablo Chong.

**Validation:** Natalia Veronica Mendoza, Pablo Chong.

**Writing – original draft:** Natalia Veronica Mendoza, Pablo Chong.

**Writing – review & editing:** Natalia Veronica Mendoza, Freddy Magdama, Nina Bogdanchikova, Pablo Chong.

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
