## [Editor Report · Decision Letter 0]

9 Aug 2024

PONE-D-24-29951Inhibition of Fusarium oxysporum growth in banana by silver nanoparticles: in vitro and in vivo assaysPLOS ONE

Dear Dr. Chong-Aguirre,

Thank you for submitting your manuscript to PLOS ONE. After careful consideration, we feel that it has merit but does not fully meet PLOS ONE’s publication criteria as it currently stands. Therefore, we invite you to submit a revised version of the manuscript that addresses the points raised during the review process.

We look forward to receiving your revised manuscript.

Kind regards,

Adalberto Benavides-Mendoza, Ph.D.

Academic Editor

PLOS ONE

Journal Requirements:

"Vanegas, E. and Chong P., received Grant FIASA-CA-2023-008 from FIASA - Fondo de Investigación para la Agrobiodiversidad, semillas y Agricultura Sustentable", research. https://www.iniap.gob.ec/el-fiasa-un-mecanismo-financiero-para-el-desarrollo-e-innovacion-de-una-agricultura-sustentable/

Bogdanchikova N. received Grant 22-13-20032 from Russian Science Foundation and Tomsk Region.

https://rscf.ru/en/news/en-57/russian-science-foundation-released-2023-results/"

"We thank “FIASA - Fondo de Investigación para la Agrobiodiversidad, semillas y Agricultura Sustentable" for funding the research project (FIASA-CA-2023-008), ESPOL Polytechnic University (ESPOL), Vicerrectorado de Investigación de la Universidad de Cuenca (VIUC), The International Biotechnology Network supported by the “Consejo Nacional de Ciencia y Tecnología”, CONACYT, Mexico, and Russian Science Foundation and Tomsk Region Grant 22-13-20032, for funding this research."

Please note that funding information should not appear in the Acknowledgments section or other areas of your manuscript. We will only publish funding information present in the Funding Statement section of the online submission form. Please remove any funding-related text from the manuscript. 

4. Please upload a copy of Figures 1 to 6, to which you refer in your text on pages 7 to 10. If the figure is no longer to be included as part of the submission please remove all reference to it within the text.

**Additional Editor Comments:**

The Figures mentioned in the main text were not included in the files sent for review.

---

## [Author Response · Author response to Decision Letter 1]

20 Aug 2024

Adalberto Benavides-Mendoza, Ph.D.,

Academic Editor,

PLOS ONE

We sincerely thank you for your thoughtful consideration and feedback on our manuscript. We appreciate the time and effort you have taken to review our work. We hope that the revisions we have made address the editors' concerns. Besides, all relevant data (raw data) of the in vitro and in vivo assays has been shared as a supporting information compressed file. We are open to any further changes that may be required. Thank you for your time and consideration.

---

## [Decision Letter · Decision Letter 1]

15 Sep 2024

PONE-D-24-29951R1Inhibition of Fusarium oxysporum growth in banana by silver nanoparticles: in vitro and in vivo assaysPLOS ONE

Dear Dr. Chong-Aguirre,

Thank you for submitting your manuscript to PLOS ONE. After careful consideration, we feel that it has merit but does not fully meet PLOS ONE’s publication criteria as it currently stands. Therefore, we invite you to submit a revised version of the manuscript that addresses the points raised during the review process.

The authors have made improvements to the manuscript. However, it is necessary to review and correct or respond to the details pointed out by the reviewers, mainly regarding the location and organization of the information presented in the different sections of the manuscript.

We look forward to receiving your revised manuscript.

Kind regards,

Adalberto Benavides-Mendoza, Ph.D.

Academic Editor

PLOS ONE

Journal Requirements:

Additional Editor Comments (if provided):

The authors have made improvements to the manuscript. However, it is necessary to review and correct or respond to the details pointed out by the reviewers, mainly regarding the location and organization of the information presented in the different sections of the manuscript.

Reviewers' comments:

Reviewer's Responses to Questions

**Comments to the Author**

1. If the authors have adequately addressed your comments raised in a previous round of review and you feel that this manuscript is now acceptable for publication, you may indicate that here to bypass the “Comments to the Author” section, enter your conflict of interest statement in the “Confidential to Editor” section, and submit your "Accept" recommendation.

Reviewer #1: (No Response)

Reviewer #2: (No Response)

2. Is the manuscript technically sound, and do the data support the conclusions?

Reviewer #1: Partly

Reviewer #2: Partly

3. Has the statistical analysis been performed appropriately and rigorously? 

Reviewer #1: Yes

Reviewer #2: Yes

4. Have the authors made all data underlying the findings in their manuscript fully available?

Reviewer #1: Yes

Reviewer #2: Yes

5. Is the manuscript presented in an intelligible fashion and written in standard English?

Reviewer #1: Yes

Reviewer #2: No

6. Review Comments to the Author

Reviewer #1: 1. The last paragraph in the Introduction Section should be the objective of this study.

2. What are the specific VCGs of those Race 1 used in this experiment?

3. How many plants per replicate used in each treatment?

4. Linking to AgNPs mode of action, any measurements to provide a direct proof?

5. How about AgNPs physiological effects on banana plants? Any plant growth parameter measurements after 40 days experiment?

6. Add biosatefty issue of AgNPs in agricultural application

Reviewer #2: Dear Authors. This is an innovative research to show alternative options to face Fusarium wilt. The found data gives the apportunity of making a very impresive paper. Howevre, he division between the sections (introductin, MM, Results...) must be standed. In some cases the sections are combined ahd there is a lack in discussion about the novelty of the results and their practical applications.

7. PLOS authors have the option to publish the peer review history of their article (what does this mean? ). If published, this will include your full peer review and any attached files.

**Do you want your identity to be public for this peer review?** For information about this choice, including consent withdrawal, please see our Privacy Policy .

Reviewer #1: **Yes: ** Sijun Zheng

Reviewer #2: No

---

## [Author Response · Author response to Decision Letter 2]

21 Oct 2024

We would like to express our gratitude to you and the reviewers for the constructive and insightful comments on our manuscript, titled “Inhibition of Fusarium oxysporum growth in banana by silver nanoparticles: in vitro and in vivo assays". We have carefully considered all the feedback and revised the manuscript accordingly.We hope that the revised version meets the expectations of the reviewers and yourself. Please do not hesitate to contact us if further clarifications are needed.

Thank you for considering our manuscript for publication in PLOS ONE.

---

## [Decision Letter · Decision Letter 2]

27 Jan 2025

Inhibition of Fusarium oxysporum growth in banana by silver nanoparticles: in vitro and in vivo assays

PONE-D-24-29951R2

Dear Dr. Chong-Aguirre,

We’re pleased to inform you that your manuscript has been judged scientifically suitable for publication and will be formally accepted for publication once it meets all outstanding technical requirements.

Kind regards,

Adalberto Benavides-Mendoza, Ph.D.

Academic Editor

PLOS ONE

Additional Editor Comments (optional):

The authors have made all the suggested adjustments, and according to the editor's review, the manuscript can be accepted for publication.

Reviewers' comments:

Reviewer's Responses to Questions

**Comments to the Author**

1. If the authors have adequately addressed your comments raised in a previous round of review and you feel that this manuscript is now acceptable for publication, you may indicate that here to bypass the “Comments to the Author” section, enter your conflict of interest statement in the “Confidential to Editor” section, and submit your "Accept" recommendation.

Reviewer #1: All comments have been addressed

2. Is the manuscript technically sound, and do the data support the conclusions?

Reviewer #1: Yes

3. Has the statistical analysis been performed appropriately and rigorously? 

Reviewer #1: Yes

4. Have the authors made all data underlying the findings in their manuscript fully available?

Reviewer #1: Yes

5. Is the manuscript presented in an intelligible fashion and written in standard English?

Reviewer #1: Yes

6. Review Comments to the Author

Reviewer #1: Suggestions to utilize current results derived from in vitro and in vivo assays to have a field trial in next step research.

7. PLOS authors have the option to publish the peer review history of their article (what does this mean? ). If published, this will include your full peer review and any attached files.

**Do you want your identity to be public for this peer review?** For information about this choice, including consent withdrawal, please see our Privacy Policy .

Reviewer #1: No

---

## [Editor Report · Acceptance letter]

PONE-D-24-29951R2

PLOS ONE

Dear Dr. Chong-Aguirre,

I'm pleased to inform you that your manuscript has been deemed suitable for publication in PLOS ONE. Congratulations! Your manuscript is now being handed over to our production team.

Kind regards,

on behalf of

Dr. Adalberto Benavides-Mendoza

Academic Editor

PLOS ONE